# Tuning Plasmonic Properties of Gold Nanoparticles by Employing Nanoscale DNA Hydrogel Scaffolds

**DOI:** 10.3390/bios13010020

**Published:** 2022-12-24

**Authors:** Mohzibudin Z. Quazi, Taeyoung Kim, Jinhwan Yang, Nokyoung Park

**Affiliations:** Department of Chemistry and the Natural Science Research Institute, Myongji University, 116 Myongji-ro, Yongin 449-728, Gyeonggi-do, Republic of Korea

**Keywords:** biosensors, polymers, nanohydrogel scaffolds, surface plasmon tuning, diagnostics, disease detection

## Abstract

Noble metals have always fascinated researchers due to their feasible and facile approach to plasmonics. Especially the extensive utilization of gold (Au) has been found in biomedical engineering, microelectronics, and catalysis. Surface plasmonic resonance (SPR) sensors are achievable by employing plasmonic nanoparticles. The past decades have seen colossal advancement in noble metal nanoparticle research. Surface plasmonic biosensors are advanced in terms of sensing accuracy and detection limit. Likewise, gold nanoparticles (AuNPs) have been widely used to develop distinct biosensors for molecular diagnosis. DNA nanotechnology facilitates advanced nanostructure having unique properties that contribute vastly to clinical therapeutics. The critical element for absolute control of materials at the nanoscale is the engineering of optical and plasmonic characteristics of the polymeric and metallic nanostructure. Correspondingly, AuNP’s vivid intense color expressions are dependent on their size, shape, and compositions, which implies their strong influence on tuning the plasmonic properties. These plasmonic properties of AuNPs have vastly exerted the biosensing and molecular diagnosis applications without any hazardous effects. Here, we have designed nanoscale X-DNA-based Dgel scaffolds utilized for tuning the plasmonic properties of AuNPs. The DNA nanohydrogel (Dgel) scaffolds engineered with three different X-DNAs of distinct numbers of base pairs were applied. We have designed X-DNA base pair-controlled size-varied Dgel scaffolds and molar ratio-based nano assemblies to tune the plasmonic properties of AuNPs. The nanoscale DNA hydrogel’s negatively charged scaffold facilitates quaternary ammonium ligand-modified positively charged AuNPs to flocculate around due to electrostatic charge attractions. Overall, our study demonstrates that by altering the DNA hydrogel scaffolds and the physical properties of the nanoscale hydrogel matrix, the SPR properties can be modulated. This approach could potentially benefit in monitoring diverse therapeutic biomolecules.

## 1. Introduction

Gold nanoparticles (AuNPs) have been vastly studied due to their facile synthesis, easy surface modulation, and strong tunable optical properties [1,2]. Gold nanoparticles have been studied broadly in preceding studies and are considered highly biocompatible [3,4]. Plasmonic nanoparticles set themselves apart from other nanoplatforms like semiconductors, quantum dots, magnetic and polymeric nanoparticles by their distinctive localized surface plasmon resonance (LSPR) [5]. Plasmonic structure construction can be exploited by precisely numbered and well-defined metal nanoparticles [6]. AuNPs have been enormously utilized due to the plasmon resonance’s high intensity, sensitivity to the particle environment, and interparticle couplings [7,8]. Plasmon resonance is an optical aid tool for examining the particles’ characteristics [9]. The SPR that results from photon confinement to a small particle size improves all the nanoparticle’s radiative and nonradiative characteristics [10,11,12]. Earlier, mainly four studies have been performed, corresponding to gold nanoparticles’ role in biosensors. LSPR, SERS, Fluorescent enhancement, and colorimetry change based on the coupling of AuNPs and hereby caused plasmonic quenching [13,14], whereas in AuNPs, a robust color change and high molar extinction coefficient can be observed based on the AuNPs state, resulting in demonstrating a naked-eye-based biosensor [15]. The size modulations and changes in the dimensions of the material result in a change in the electronic properties. The density of state and the spatial and length scale of electronic motion oscillate drastically as size varies [16]. Despite having been studied for the past few decades, utilization and construction of nanoscale-based biomaterials to tune the SPR properties are still challenging. The cytotoxicities and uncontrolled interparticle distance while assembly of noble metals with protein, DNA, and surface ligands to demonstrate coupled plasmonic properties are still arduous. Here, we put forth the design of nanoscale Dgel scaffolds with the magnitude of electrostatic attraction Dgel:AuNPs nano-assemblies for tuning AuNP’s plasmonic properties. In this study, we hypothesize utilizing the Dgel:AuNPs nano-assembly as biosensors. The known 3D hydrophilic hydrogel’s high biocompatibility and negligible toxicity make it feasible and could serve as a great platform for tuning the AuNP’s SPR properties [17,18,19]. Here, we used a nanoscale Dgel scaffold in tuning the plasmonic properties of AuNPs, which has a potential role comparatively in earlier studies. In preceding reports, gold nanoarrays/scaffold has been utilized, which could affect renal clearance and non-biodistributions due to the hydrodynamic sizes. Our nanoscale Dgel scaffold’s biocompatibility, easy biodegradability, and negligible cytotoxicity make it widely applicable in high biodistribution and distinctive. The hydrophilic polymer network incorporates negligible cytotoxicity and a simple mechanism. Our Dgel:AuNPs nano-assemblies could be altered by differently modified ligands for further applications in bioassay due to high surface-to-volume ratios. Similarly, the charge-driven mechanism will help to manage the interparticle distance completely based on the charge hindrance factor without causing agglomerations. Here, we report the Dgel scaffolds with different sizes of X-DNAs effectively incorporated in delivering the different sizes of Dgels. Akin X-DNA concentration change will cause the change in size thus we can effectively control the role of the X-DNA-based Dgel engineering system. Moreover, the steering of plasmonic properties of AuNPs could be effectively altered by Dgel scaffold size variations with the changes in the molar ratios of Dgel and AuNPs, respectively. Therefore, we further demonstrate the Dgel:AuNPs nano-assembly-driven and effectively controlled tuning of SPR properties in gold nanoparticles.

## 2. Experimental Section

### 2.1. Materials and Equipment

All the chemicals were used as received without further purification. hydrogen tetrachloroaurate (HAuCl_4_), sodium borohydride (NaBH_4_), sodium citrate dihydrate, magnesium chloride (MgCl_2_), Tris/borate/EDTA (TBE), Tris/acetate/EDTA (TAE), tris EDTA buffer (TE), quaternary ammonium ligands, chemicals were obtained from Sigma Aldrich. Oligonucleotide sequences were purchased from Integrated DNA Technologies (Coralville, IA, USA), as mentioned in Appendix A. T4 DNA ligase and 10× T4 DNA ligase buffer were obtained from Takara. All the experiments were carried out using triply distilled water. UV vis absorption spectrums were recorded from the Eppendorf bio spectrometer. Gel electrophoresis analyses were performed on Bio-Rad powerpack horizontal electrophoresis. Hydrodynamic size and zeta potential studies were performed using Malvern Zetasizer Nano. Scanning electron microscopic images were recorded using SU-70/HORIBA (Hitachi, Japan) at 15.0 kV. Digital images were captured using a digital camera.

### 2.2. Methods

#### 2.2.1. Engineering of X-DNA-Based Nanoscale DNA Hydrogel (Dgel) Scaffold

DNA nanohydrogel (Dgel) ligation was carried out as an earlier reported method [17] using 25, 50, 100, and 150 µM concentrations of X-DNA monomers. X-DNA monomers building blocks were mixed with 10 units of T4 DNA ligase and T4 DNA ligase buffer to perform ligation. After mixing, the solution was kept on incubation at a reaction condition of 16 °C for 12 h. All three ends of X-DNA were modified with sticky ends at 5′ and kept one end blunt at the edge to systematically control the sizes of Dgel. Afterward, several washing steps using distilled water were carried out to remove non-ligated free particles. Dgel engineering was confirmed by gel electrophoresis and DLS studies.

#### 2.2.2. Nano Assembly of X-DNA-Based Dgel Scaffolds with Plasmonic Gold Nanoparticles

Synthesis of AuNPs was followed as an earlier reported protocol [20]. A 10 mM aqueous solution of HAuCl_4_ was mixed on a stirring hot plate with 250 mL distilled water. An aqueous solution of 50 mM of sodium citrate tribasic dihydrate was added to the rapidly stirred boiling solution. A gradual color change was observed, followed by the quick addition of sodium citrate tribasic dihydrate. The color of the solution changed from yellow to purple and became wine red. Subsequently, the red-colored solution was then removed from the hot plate and allowed to cool at room temperature for 30 min. Next, dialysis was carried out by using 100 KDa Mw Amicon centrifugal filters. After the washing, the synthesized AuNPs were mixed with excessive quaternary ammonium ligands and stirred overnight at room temperature. The solution was dialyzed using the centrifugal filter to remove the excessive ligands in the solution followed by stirring. To proceed with the nano-assembly of Dgel:AuNPs, quaternary ammonium ligand-modified positively charged AuNPs were mixed with fabricated Dgels (Figure 1). Dgel:AuNPs solutions with variable molar ratios were kept on a shaker at 1000 RPM for 6 h at 16 °C. Subsequently, centrifugation and decanting steps using distilled water at 6000 RPM for 30 min were performed. Next, the nanoscale Dgel scaffold’s assemblies with AuNPs were investigated to observe the tuning of AuNPs. To assess the possible alteration in SPR properties of AuNPs, we utilized differently (concentration and base pair-based Appendix A) engineered Dgels to assemble with AuNPs at different molar ratios. Furthermore, instrumental analyses were carried out.

## 3. Result and Discussion

### 3.1. Synthesis and Characterization of Variable Base Pair Designed X-DNA-Based Dgel Scaffolds and Size-Controlled Modulations

Hydrogels, 3D- hydrophilic structured biomaterials have been widely used as substrates for plasmonic particles [20,21,22]. The mechanical and structural properties make hydrogels more sensitive, selective, efficient, and robust sensing probes. The soft material tendency of hydrogels has been recognized as possibly altering the sensing properties when used as substrates [23]. We approached synthesizing the size-controlled nanoscale-based hydrogel scaffolds on the preceding challenges and afore-discussed limitations. As aforementioned, three different base pairs varied (36, 56, and 76 bp) X-DNAs were designed. Prior to the Dgel ligation, the engineering of X-DNAs of different base pairs was confirmed by using gel electrophoresis (Figure 1A). The molecular weight differences of X-DNAs draw a gradient change in band positions. Followed by X-DNAs confirmation, a ligation reaction was performed to engineer X-DNA-based nanoscale Dgels. Figure 1B–D showcases a gel electrophoresis image of Dgel engineered with the prior mentioned X-DNA monomers at different concentrations. Figure 1B–D lanes 3–6 show that at four different concentrations (25, 50, 100, and 150 µM), the formation of nanoscale Dgels scaffold was carried out successfully. The intense band appearance at the hollow pocket of gel (well part of the band) in the gel electrophoresis image suggests the evident formation of Dgels at varied concentrations.

Furthermore, to ascertain the confirmation of Dgel scaffolds engineering DLS sizes, SEM images, and zeta potential assessments were performed (Figure 2). The DLS studies put forth our hypothesis, the formation of nanoscale Dgel scaffolds could be altered and controlled by varying the base pair numbers and concentration of X-DNA monomers. The lower the molar concentration or the higher the number of base pairs of X-DNA, the larger the size of Dgel was produced. As Figure 2A DLS demonstrates, at low concentrations of 36 bp, X-DNA fabricates larger sizes of Dgel (304.5 ± 15.0 nm), whereas, at high concentrations, the size of the same monomer fabricated Dgel shrinks to (50.5 ± 4.0 nm). Correspondingly, 56 bp and 76 bp X-DNA modified Dgels have shown similar trends in the sizes at variable concentrations. The 56 bp X-DNA modified Dgel at low concentration provided a Dgel scaffold with a hydrodynamic size of (394.0 ± 19 nm), whereas at a high concentration of 56 bp X-DNA fabricated a Dgel of the hydrodynamic size of (215.7 ± 9.5 nm). Likewise, the 76 bp X-DNA modified Dgel at low concentration engineered a Dgel scaffold with a hydrodynamic size of (406.2 ± 21.8 nm) and at a high concentration of 76 bp X-DNA fabricated A Dgel of the hydrodynamic size of (279.3 ± 12.7 nm). Therefore, here, we demonstrate that the variation in base pairs could contribute highly to controlling the size scale of engineered Dgel scaffolds. The change in concentrations could play a key role in the modulation of nanoscale Dgel scaffolds. Correspondingly, zeta potential studies assayed the negative scaffold of Dgels (Figure 2B). Likewise, the SEM image attributes indistinguishable results suggesting the size variations could be altered as aforesaid (Figure 2C). Next, to demonstrate the active role of the nanoscale Dgel scaffolds in tuning plasmonic properties, we proceed to nano assembly of Dgel:AuNPs as introduced in Figure 1.

### 3.2. Dgel Scaffold-Based Tuning of Plasmonic Properties

The UV visible analysis ascertained the electrostatic attraction felicitated nano assembly of Dgel:AuNPs. UV visible studies were executed to understand the tuning of plasmonic properties in AuNPs. In earlier reported studies, size, and shape-dependent tuning of surface plasmonic properties were carried out with different methods [24,25]. In this study, we have performed electrostatic force-driven convenient nano assembly of Dgel:AuNPs causing a plasmon shift from UV vis to near-infrared wavelength. The molar ratio calibrated Dgel:AuNPs nano assembly influenced the plasmonic properties of AuNPs. The nano assemblies of Dgel:AuNPs were assayed with alteration in molar ratios as Dgel [36 bp X-DNA]:AuNPs with 1:0.2, 1:0.4, and 1:0.6 at different concentrations of engineered Dgel each. The observed UV-visible studies demonstrate the evident and gradient plasmonic redshifts (Figure 3). We utilized the Dgel engineered for further assembly stages with variable concentrations of 36 bp X-DNAs (25, 50, 100, and 150 µM). The dielectric properties of the material and the frequency of the dipole resonance depend upon the size and shape of the nanoparticles [26]. Correspondingly, at a higher concentration of 150 µM Dgel:AuNP assembly showed the plasmonic absorption spectrum redshifted to 603 nm wavelength at a mixing ratio of Dgel [36 bp X-DNA]:AuNPs 1:0.6 (Figure 3A). Likewise, at 100 and 50 µM (Figure 3B,C) with similar mixing ratios of Dgel [36 bp X-DNA]:AuNPs, the gradient redshift happened. Referring to Figure 3, the redshifts were gradual, while in the low-concentration instances, the shifts dependent on molar ratios were drastically moved (Appendix A). In the instance of 25 µM concentration-based Dgel scaffold assembly of AuNPs, the plasmonic peak shifted moderately from 516 up to 587 nm (Figure 3D). The plasmonic shift changed drastically near to infrared along with the change in molar ratios as Dgel [36 bp X-DNA]:AuNPs with 1:0.4, (532 nm), and 1:0.6 shifted to 587 nm. The plasmonic absorption’s redshift exhibits the spectrum shift with an extended tail to the near-infrared region. Similarly, scanning electron microscope (SEM) image analysis constitutes the formation of Dgel:AuNPs nano assemblies with varied molar ratios (Appendix A). The successful demonstration of the Dgel [36 bp X-DNA]:AuNPs assemblies and plasmonic redshift allowed us to investigate the assembly of gold nanoparticles under various conditions, especially, the variable Dgel scaffold sizes with variable molar ratios of the Dgel [56 bp X-DNA]:AuNPs and Dgel [76 bp X-DNA]:AuNPs.

Next, we performed similar Dgel:AuNPs assemblies with Dgel [56 bp X-DNA] by increasing the molar ratios till the saturation points were observed. We observed the nano-assemblies of Dgel [56 bp X-DNA]:AuNPs with the change of molar ratio from 1:0.2 to 1:1.2, and 1: 1.4 (Figure 4). Similarly, the trends of redshifts in plasmonic absorptions were observed. The Dgel:AuNPs nano-assemblies caused redshifts that are correlative to the increased molar ratios in Dgel [56 bp X-DNA]:AuNPs (Appendix A). The redshift in Dgel [56 bp X-DNA] established scaffolds interpret the far redshift trend at different mixing ratios of Dgel [56 bp X-DNA]:AuNPs. As in the 150 µM Dgel scaffold instance, the ratios varied from 1:0.2 (529 nm) to 1:1.0 (589 nm), whereas, in the case of 25 µM Dgel scaffold the alteration in plasmonic absorption shifted from 1:0.2 (529 nm) to 1:1.4 (548 nm). These observations speculate the Dgel scaffolds of varied concentrations felicitated AuNPs to flocculate in increasing trends. As more AuNPs are assembled to scaffolds, this signifies the strongly coupled plasmon modes. The nano-assembly equally contributed to maintaining the interparticle distances per se. We have observed in the SEM image analysis the Dgel:AuNPs nano assemblies at varied molar ratios contemplating the earlier-mentioned observations (Appendix A). Furthermore, we observed similar trends and redshifts with plasmonic properties alterations in Dgel scaffolds engineered with [76 bp X-DNA], the variation and change in mixing of molar ratio resulted in the alteration of plasmonic properties and redshifts (Figure 5 and Appendix A). Similarly, SEM image observation of varied molar ratio Dgel:AuNPs nano assemblies showcase the varied Dgel:AuNPs nano assemblies (Appendix A), whereas the trendline shift in UV visible studies elucidates our objective, the plasmonic properties could be successfully tuned by utilizing Dgel scaffolds of altered sizes and concentrations. These instances could be employed as a key role in tuning SPR of noble metals with different molar mixing ratios, likewise, changes in Dgel concentrations.

In the analysis of SEM, it is speculated that Dgel [36 bp X-DNA]:AuNPs assembly with change in molar ratios can contribute to tuning the plasmonic properties. The size change of Dgel:AuNPs could accelerate the interparticle-coupled plasmons and result in plasmonic shifts (Figure 6 and Appendix A). Moreover, the assembly of AuNPs could be systemically controlled by modulating the molar ratio of Dgel:AuNPs. The molar ratio-based nano assembly of Dgel:AuNPs distinctly exerts the role of the Dgel scaffold can modulate the blue shift from UV visible region to NIR. The broad shift of the visible spectrum band to the NIR region corresponds to electron oscillations which could be tuned by Dgel scaffold alteration based on our obtained results. Size confinement provides unique electronic and optical properties. Similarly, the strong color of noble metal nanoparticle colloidal solutions, caused by surface plasmon absorption, is a distinguishing feature of noble metal nanoparticles. The red shift in the peaks of Dgel:AuNPs nano-assemblies determines the evident change in the SPR of AuNPs with the naked eye (Appendix A). We observed a similar change in the color of AuNPs obtained after Dgel:AuNPs nano assembly at varied molar ratios. These digital images confirm the plasmonic shift and indicate the appearance of strong coupled plasmon modes. The solution’s color changes from pink to deep purple into deep violet, drawing attention to Dgel scaffold-mediated AuNPs nano assemblies and size confinement by assemblies. These nano assemblies’ absorbance in the far-red and near-infrared by scaffold influence suggests that they could be employed as a potential probe in deep tissue penetration therapeutics [26].

## 4. Conclusions

Here, we demonstrated a feasible approach for the tuning of plasmonic properties of AuNPs. We utilized nanoscale Dgel as a scaffold to finely tune the SPR properties. The electrostatic force could be the potential approach to assemble nanoparticles on non-hazardous biomaterials. We investigated the role of the Dgel scaffold’s concentration-based alteration in the SPR properties. We have successfully employed the Dgel scaffold with variable X-DNA base pair sizes. By controlling the amount of Dgel:AuNPs molar ratios the SPR of AuNPs could be tuned to the NIR region with a specified wavelength. Similarly, based on the observed results, Dgel:AuNPs nano-assembly could be shifted for desired properties by controlling Dgel as a scaffold. The total light extinction of nano-assembled Dgel:AuNPs with SPR around 800 nm is dominated by absorption that describes them as a suitable agent for photothermal therapies. This platform could be potentially used for cancer imaging, SERS imaging, and SPR incorporating biosensors etc.

## Data Availability

For further correspondences and data availability please contact to pospnk@mju.ac.kr.

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
