# Peer review of "Tuning Plasmonic Properties of Gold Nanoparticles by Employing Nanoscale DNA Hydrogel Scaffolds"

_biosensors, 2022, doi:10.3390/bios13010020_

Round 1
Reviewer 1 Report
The manuscript “Tuning plasmonic properties of gold nanoparticles by employing nanoscale DNA hydrogel scaffolds” developed a X-DNA-based Dgel. The authors demonstrated meticulous characterizations of such system.
The work is interesting and precedent, although it is preliminary.
I think it is acceptable after some revision, taking into account the following points.
Major points:
1. The content presented in this paper does not align well with this special issue of Biosensors. From the website, https://www.mdpi.com/journal/biosensors/special_issues/tren_poly_bio “This Special Issue will focus on the fabrication and application of various biosensors consisting of functional polymer platforms which realize a versatile implementation of sensors toward in vitro as well as in vivo diagnostics for healthcare, disease detection, environmental monitoring, and other related fields.” The authors demonstrated that the SPR properties could be modulated, yet not sensing ability of the system. To be accepted into this issue, the authors need to demonstrate the sensing ability of such system. Otherwise, a materials-focused journal might be a better option for this paper.
Minor points:
1. Sample size is missing from data presentation. Figure 2 should fully describe data presentation with sufficient details (e.g., Mean ± SD). What is the sample size in Figures?
2. Page 6. “36 bp X-DNAs (25 uM, 50 uM, 100 uM, and 150 uM).” Unit of the concentrations should be presented as μM, not uM. Please check and correct throughout both main text, and SI (for example, Figure S1).
3. Figure 6. Scale bars are missing for figure B, C, D. Additionally, for this multi-panel figure, only panel A is labeled, please label “B”, “C”, “D” for SEM images in the figure.
Author Response
Dear Reviewer,
Thank you so much for the comments on our manuscript and We appreciate all the sincere comments by you. The comments were carefully considered during the revision of the manuscript. In the upcoming pages, we tried to respond to these comments one by one. Our detailed list of changes and responses to your concerns are given below.
Response to Reviewer 1
Comments 1: The content presented in this paper does not align well with this special issue of Biosensors. From the website, https://www.mdpi.com/journal/biosensors/special_issues/tren_poly_bio “This Special Issue will focus on the fabrication and application of various biosensors consisting of functional polymer platforms which realize a versatile implementation of sensors toward in vitro as well as in vivo diagnostics for healthcare, disease detection, environmental monitoring, and other related fields.” The authors demonstrated that the SPR properties could be modulated, yet not sensing ability of the system. To be accepted into this issue, the authors need to demonstrate the sensing ability of such system. Otherwise, a materials-focused journal might be a better option for this paper.
|
Our response: We thank the reviewer for the helpful comment.
Following to reviewer’s valuable comment, we would like to add a few points to our engineered scaffold of DNA nanohydrogel and gold nanoparticles (AuNPs) nano assemblies. The purpose of the utilization of AuNPs was to enact as a biosensor. Referring to preceding studies utilization of AuNP’s plasmonic properties makes a fortune impact as a biosensor. Reference no. 2 in the manuscript elucidates the employment of AuNP’s plasmonic shift in biosensors. Similarly, AuNPs plasmonic tuning could provide optical signal enhancement to exert a major role as a biosensor in Raman imaging and Raman scattering. Whereas AuNP’s distinctive properties and their excellent biocompatibility, conductivity, and high surface-to-volume ratio support as a widely used element in the field of bioassay”. Moreover, in the manuscript, we wanted to highlight that biosensing employment could be utilized by modulating the surface of AuNPs and DNA hydrogel due to the high feasibility of surface alterations. We believe our engineered Dgel-AuNPs nano assemblies would contribute at the same scale.
Reference 1 – (DOI: https://doi.org/10.3390/ijms19072021 )
Reference 2 – (DOI: https://doi.org/10.3390/nano8120977 )
Additionally, we would like to bring the reviewer’s attention to our previously published article, https://doi.org/10.1021/acsabm.1c00946. In this article, we utilized AuNPs for reversible pH-responsive-based aggregation in the intracellular microenvironment for cell imaging purposes. The Raman imaging and Raman mapping strategies were employed with the AuNPs as the only probe in the cancer cell. We kindly hope through these preceding studies and references it can be inferred that our engineered complex could be employed as a biosensor as well as could contribute to biosensing applications. Therefore, we desire to be eligible for this special issue publication.
Comments 2: Sample size is missing from data presentation. Figure 2 should fully describe data presentation with sufficient details (e.g., Mean ± SD). What is the sample size in Figures? |
Our response: The reviewer’s point is well taken. we have added the sample sizes with a concise description of the Hydrodynamic mean sizes. The sample sizes, and mean ± SD are included based on the three replicates sets of experiments.
We have added the concise description in our revised manuscript (p4 line 7 & line 8) and (p5 lines 2 to 7)
Comments 3: Page 6. “36 bp X-DNAs (25 uM, 50 uM, 100 uM, and 150 uM).” Unit of the concentrations should be presented as μM, not uM. Please check and correct throughout both main text, and SI (for example, Figure S1). |
Our response: We thank the reviewer for the helpful comment. We have thoroughly reviewed the unit of the concentrations and the reviewer’s comment is well considered. We regret the inconvenience caused.
Comments 4: Figure 6. Scale bars are missing for figure B, C, D. Additionally, for this multi-panel figure, only panel A is labeled, please label “B”, “C”, “D” for SEM images in the figure. |
Our response: We sincerely thank the reviewer for the helpful comment. We have added the scale bars. Next, we have added a detailed description of the SEM images corresponding to the labeled panels.
Reviewer 2 Report
The manuscript entitled “Tuning Plasmonic Properties of Gold Nanoparticles by Employing Nanoscale DNA Hydrogel Scaffolds”. It presents a current approach and the results are very interesting both for academic research areas and for biomedical engineering, microelectronics and even for catalysis. The manuscript reads very well with interesting and practice-oriented results from monitoring various therapeutic biomolecules. Characterization techniques provided insightful information about title synthesis. It is recommended that this manuscript be published after including and addressing the comments listed below with minor comments:
It would be possible to clearly explain the innovation and importance of work on the introduction of the manuscript, justifying the value of the work and comparing it to similar previously published work, justifying the value of the work and comparing it to similar previously published work. They should develop the advantage and potential applications that would benefit from monitoring diverse therapeutic biomolecules.
It is suggested that SEM image scale bars be added and clearly presented in the revised manuscript.
Although the manuscript provided some characterization techniques for determining physicochemical properties of AuNPs. The manuscript could benefit from the addition of additional methods, such as X-ray photoelectron spectroscopy for measuring metal oxidation states, as well as EDS, FTIR, and TEM analyzes for the various species discussed.
Please correct in the revised manuscript the units ml to mL.
Finally, I suggest that the tables and figures mentioned in the Supplementary be added in the revised manuscript. Both the readers and the manuscript will benefit from the addition of tables and figures.

Author Response
Response to Reviewer2:
Title: Tuning Plasmonic Properties of Gold Nanoparticles by Employing Nanoscale DNA Hydrogel Scaffolds
Dear Reviewer,
Thank you so much for the comments on our manuscript and We appreciate all the sincere comments by you. The comments were carefully considered during the revision of the manuscript. In the upcoming pages, we tried to respond to these comments one by one. Our detailed list of changes and responses to your concerns are given below.
Response to reviewer 2
Comments 1: It would be possible to clearly explain the innovation and importance of work on the introduction of the manuscript, justifying the value of the work and comparing it to similar previously published work, justifying the value of the work and comparing it to similar previously published work. They should develop the advantage and potential applications that would benefit from monitoring diverse therapeutic biomolecules. |
Our response: We thank the reviewer for the helpful comment. We have thoroughly swotted the introduction part in our revised manuscript. We believe our revised manuscript has been much improved. Following to reviewer’s valuable comment, we have added a few points for the required justifications and tried to deliver some comparative points in response to preceding studies.
We added points to our revised manuscript (p2 line 17~27).
Comments 2: It is suggested that SEM image scale bars be added and clearly presented in the revised manuscript
|
Our response: The reviewer’s comment is well taken. The SEM image scale bars were added to the corresponding figures in the revised manuscript.
Comments 3: Although the manuscript provided some characterization techniques for determining physicochemical properties of AuNPs. The manuscript could benefit from the addition of additional methods, such as X-ray photoelectron spectroscopy for measuring metal oxidation states, as well as EDS, FTIR, and TEM analyzes for the various species discussed.
|
Our response: We thank the reviewer’s valuable comment. We would like to add a few points in response to the reviewer’s comments. Our motto while executing this design was to discover and highlight the role of nanoscale Dgel scaffolds influencing the tuning of the AuNP’s SPR properties. The alteration of sizes in X-DNA monomers and their concentration could alter the SPR properties of AuNPs. We would like to bring the reviewer’s attention, as previous researchers have done high-scale studies on the physiochemical properties of AuNPs that are widely known. Akin, we followed a similar synthesis method of AuNPs in this manuscript as earlier reported. We have cited several references from authors such as “Mostafa El-Sayed et al” to provide detailed required referral studies about AuNPs. Therefore, we wanted to highlight and present the tuning properties of AuNPs that are highly influenced by the concentration and molar ratios of differently engineered Dgels.
We believe the manuscript’s focus on the Dgel scaffold contributes at a major scale. Simultaneously “Dgel: AuNPs nano assemblies caused the tuning of plasmonic properties” focus will shift and to avoid this we used the essential physicochemical properties only. We would highly appreciate the reviewer’s kind consideration.
Comments 4: Please correct in the revised manuscript the units ml to mL.
|
Our response: We sincerely thank the reviewer for the helpful comment, here we apologize for the error. In the revised manuscript we swotted the issue.
Comments 5: Finally, I suggest that the tables and figures mentioned in the Supplementary be added in the revised manuscript. Both the readers and the manuscript will benefit from the addition of tables and figures. |
Our response: We thank the reviewer for the helpful comment. We would like to respond humbly that we wanted to keep the manuscript concise and simultaneously we have cited the relevant supplementary figures and the required details to the corresponding context in order to provide feasible reading for the readers. We sincerely thank the reviewer’s kind suggestion.
Reviewer 3 Report
In this MS, authors claimed they made different Dgel scaffolds to assemble AuNPs arrays with different SPR bands. However, this work is a primary work without real application investigation.
Additionally
1. X-DNA morphologies lack direct experimental evidences.
2. The morphology observation on Dgel scaffolds made by using 56 and 76 bp X-DNA is missing.
3. In Figure 2, with the concentration changing, the Zeta potential changes but no reasonable interpretation is given in the main context.
4. In Figure 6, the scale bars are missing for SEM images. For the convincing, the enlarged SEM views for AuNPs assembly formed by using different ratios should be provided.
5. The preparation reproducibility and storage stability of such AuNPs assemblies on the Dgel scaffolds must be checked.
6. An abbreviation occurring in the first place in abstract and in main context should be followed by full name.
7. All the experiments were carried out using triply distilled water. However, in experimental process, DI water was used.
8. In abstract, the term of felicitates is typo.
Author Response
Response to Reviewer3:
Title: Tuning Plasmonic Properties of Gold Nanoparticles by Employing Nanoscale DNA Hydrogel Scaffolds
Dear Reviewer,
Thank you so much for the comments on our manuscript and We appreciate all the sincere comments by you. The comments were carefully considered during the revision of the manuscript. In the upcoming pages, we tried to respond to these comments one by one. Our detailed list of changes and responses to your concerns are given below.
Response to reviewer 3
Comments 1: X-DNA morphologies lack direct experimental evidences. |
Our response: We thank the reviewer for the valuable comment, we would like to bring the reviewer’s attention to figure 1. We have performed the gel electrophoresis analysis for the confirmation of X-DNA formation at varied base pairs. The electrophoresis analysis and our previous reference work allowed us to proceed with further Dgel engineering studies.
Moreover, we would like to refer to our previous studies on X-DNA and Dgel engineering from which we inspire to program X-DNA utilization for SPR tuning properties. We have followed the strategies and protocols akin to studies while making X-DNA and DNA nanohydrogels (Dgel).
- https://doi.org/10.1038/nmat1741
- https://doi.org/10.1038/nmat2419
- https://doi.org/10.1038/s41467-018-06864-0
- https://doi.org/10.1039/C6RA03810G
Comments 2: The morphology observation on Dgel scaffolds made by using 56 and 76 bp X-DNA is missing. |
Our response: We thank the reviewer’s valuable comment. The Dgel scaffold morphologies from 56 & 76 bp X-DNAs were confirmed by gel electrophoresis mobilities. Secondly, the SEM images of only the Dgel scaffold could be ambiguous due to the nature of the analysis, we preferred to provide UV visible and SEM image analysis of Dgel: AuNPs nano assemblies. We have added the SEM observed images of Dgel: AuNPs nano assemblies at varied ratios in the supplementary information pp. 5,7 & 9 (Figure S2, S4 & S6). Moreover, SEM image analysis and the scaffold’s short briefing have been mentioned to put forth the observed results. The UV visible studies of 56bp and 76bp X-DNA engineered Dgel scaffolds in Figures 4 & 5 have also contributed similarly to the evident observation on the engineered Dgel scaffold and Dgel: AuNPs nano assemblies.
Comments 3: In Figure 2, with the concentration changing, the Zeta potential changes but no reasonable interpretation is given in the main context. |
Our response: We thank the reviewer’s valuable comment. The intent for the zeta potential studies was only to assure the engineered Dgel scaffolds stand negatively charged consistently. The observed zeta potential of differently engineered Dgel scaffolds interprets the corresponding negative scaffolds. The consistent negatively charged scaffolds were suitable for the electrostatic attraction-based nano assemblies. Thus, from the observed studies, we proceed with further AuNP’s nano assemblies as Dgel: AuNPs.
Comments 4: In Figure 6, the scale bars are missing for SEM images. For the convincing, the enlarged SEM views for AuNPs assembly formed by using different ratios should be provided. |
Our response: The reviewer’s comment is well taken. Following the reviewer’s suggestion, we have added an enlarged SEM image of the Dgel: AuNPs nano assembly formed by varied molar ratios of Dgel: AuNPs in the supplementary information p5 (Figure S2).
Comments 5: The preparation reproducibility and storage stability of such AuNPs assemblies on the Dgel scaffolds must be checked. |
Our response: We thank the reviewers for the comment. The observed result has been introduced in the manuscript followed by three replicate of the experiments. The scaffolds reproducibilities have not observed any hurdles meanwhile the experiments were carried out. Similarly, for stability and storage, we observed the Dgel: AuNPs nano assemblies were highly stable for more than a month.
Comments 6: An abbreviation occurring in the first place in abstract and in main context should be followed by full name. |
Our response: We sincerely thank the reviewer for the valuable comment. We have followed the reviewer’s suggestion.
Comments 7: All the experiments were carried out using triply distilled water. However, in experimental process, DI water was used.
|
Our response: We thank the reviewer for the helpful comment. All the experiments were carried out using triply distilled water. In the revised manuscript we reviewed our mistake; we apologize for the inconvenience caused.
Comments 8: In abstract, the term of felicitates is typo.
|
Our response: We thank the reviewer for the helpful comment. Following the reviewer’s suggestion, we reviewed the typing mistake.
Round 2
Reviewer 1 Report
The manuscript “Tuning plasmonic properties of gold nanoparticles by employing nanoscale DNA hydrogel scaffolds” developed a X-DNA-based Dgel. The authors demonstrated meticulous characterizations of such system.
The authors have suitably edited the manuscript, addressing the reviewer comments and clarifying key elements.
I believe that the manuscript now should be published.
Reviewer 3 Report
The revision could be accepted for publication as it is.